# Use of medicinal plants for COVID-19 prevention and respiratory symptom treatment during the pandemic in Cusco, Peru: A cross-sectional survey

**Magaly Villena-Tejada**[1]*, **Ingrid Vera-Ferchau**[1], **Anahí Cardona-Rivero**[1], **Rina Zamalloa-Cornejo**[2], **Maritza Quispe-Florez**[3], **Zany Frisancho-Triveño**[1], **Rosario C. Abarca-Meléndez**[4], **Susan G. Alvarez-Sucari**[4], **Christian R. Mejia**[5], **Jaime A. Yañez**[6,7]*

1 Departamento Académico de Farmacia, Facultad de Ciencias de la Salud, Universidad Nacional de San Antonio Abad del Cusco, Cusco, Peru, 2 Departamento Académico de Matemáticas y Estadística, Facultad de Ciencias, Universidad Nacional de San Antonio Abad del Cusco, Cusco, Peru, 3 Departamento Académico de Biología, Facultad de Ciencias, Universidad Nacional de San Antonio Abad del Cusco, Cusco, Peru, 4 Escuela Profesional de Farmacia y Bioquímica, Facultad de Ciencias de la Salud, Universidad Nacional de San Antonio Abad del Cusco, Cusco, Peru, 5 Facultad de Medicina, Universidad Continental, Lima, Peru, 6 Vicerrectorado de Investigación, Universidad Norbert Wiener, Lima, Peru, 7 Gerencia Corporativa de Asuntos Científicos y Regulatorios, Teoma Global, Lima, Peru

* magaly.villena@unsaac.edu.pe (MVT); jaime.yanez@uwiener.edu.pe (JAY)

**Data Availability Statement:** Anonymized data set supporting the findings of this study is stored at

## Abstract

### Background

The burden of the COVID-19 pandemic in Peru has led to people seeking alternative treatments as preventives and treatment options such as medicinal plants. This study aimed to assess factors associated with the use of medicinal plants as preventive or treatment of respiratory symptom related to COVID-19 during the pandemic in Cusco, Peru.

### Method

A web-based cross-sectional study was conducted on general public (20- to 70-year-old) from August 31 to September 20, 2020. Data were collected using a structured questionnaire via Google Forms, it consisted of an 11-item questionnaire that was developed and validated by expert judgment using Aiken's V (Aiken's V > 0.9). Both descriptive statistics and bivariate followed by multivariable logistic regression analyses were conducted to assess factors associated with the use of medicinal plants for COVID-19 prevention and respiratory symptom treatment during the pandemic. Prevalence ratios (PR) with 95% Confidence Interval (CI), and a P-value of 0.05 was used to determine statistical significance.

### Results

A total of 1,747 respondents participated in the study, 80.2% reported that they used medicinal plants as preventives, while 71% reported that they used them to treat respiratory symptoms. At least, 24% of respondents used medicinal plants when presenting with two or more respiratory symptoms, while at least 11% used plants for malaise. For treatment or

the Dryad data repository (https://datadryad.org/stash/share/Yke7zt5MuVeD7aE8ie5G_jrbYPE8ZaRCLH58FuYI9QI).

**Funding:** The authors would like to thank the Universidad Nacional de San Antonio Abad del Cusco (UNSAAC) grant R-446-2020-UNSAAC. The funder provided support in the form of expenses related to the data collection from the survey, commercial license for the statistical software, translation and publication fees. The funder did not have any additional role in the study design, data collection and analysis, decision to publish, or preparation of the manuscript. The specific roles of these authors are articulated in the Author Contributions section.

**Competing interests:** The authors declare that they have no conflict of interests. JAY is currently employed by the commercial company Teoma Global, but there is no conflict of interest related to any of the medicinal plants or commercial products. This does not alter our adherence to PLOS ONE policies on sharing data and materials.

prevention, the multivariate analysis showed that most respondents used eucalyptus (p < 0.001 for both), ginger (p < 0.022 for both), spiked pepper (p < 0.003 for both), garlic (p = 0.023 for prevention), and chamomile (p = 0.011 for treatment). The respondents with COVID-19 (p < 0.001), at older ages (p = 0.046), and with a family member or friend who had COVID-19 (p < 0.001) used more plants for prevention. However, the respondents with technical or higher education used less plants for treatment (p < 0.001).

## Conclusion

There was a significant use of medicinal plants for both prevention and treatment, which was associated with several population characteristics and whether respondents had COVID-19.

## Introduction

COVID-19 was declared a pandemic by the World Health Organization (WHO) on March 11, 2020 because of its rapid transmission and infection rates worldwide [1, 2]. This disease is characterized by a progressive and severe pneumonia, and the most common symptoms are fever, dyspnea, dry cough, fatigue, headache, anosmia, and ageusia [2–4]. However, recent evidence indicates that multiple neurological complications, besides anosmia, could present in COVID-19 patients [5]. Some of those neurological complications include headache, myalgia, dizziness, encephalitis, stroke, epileptic seizures and Guillain-Barre syndrome [5]. As of May 11, 2021, more than 159 million global confirmed cases and more than 3 million deaths have been reported [6]. The first confirmed case in Peru was reported on March 8, 2020 [7], and the number of cases rapidly increased despite the measures established by the Peruvian government [8, 9]. In less than four months, Peru ranked second among Latin American countries, following Brazil, for the highest number of COVID-19 cases and deaths [10, 11]. Physical isolation was the main preventive measure implemented worldwide to avoid the contagion [8, 12, 13], which caused multiple lifestyle changes in people. Many people have experienced the death of family and friends [14–17], which has resulted in anxiety and mental distress [18–21]. The widespread disinformation [7, 22], fake news [9] and anti-vaccine comments [23, 24] have caused an increase in self-medication [25], use of medicinal plants, and other alternative treatments [26]. Many have urged that the general state of disinformation be addressed by governmental institutions [7, 27, 28]. Multiple publications have illustrated the fragmented healthcare system in Peru, which has not been the most effective during the COVID-19 pandemic resulting in a high number of physicians' deaths [29], limited public policies [30], and detrimental effects in the mental status of the population [21, 27]. Furthermore, Peru has reported discrepancies in the official reports of COVID-19 deaths nationwide [31], poor execution of SARS-CoV-2 testing and reporting [17], and an increase number of COVID-19 cases in children and adolescents [8, 16].

The current pandemic generates fear in the population who seek solutions to prevent or alleviate the symptoms of the disease since they feel the only resource available to the is to self-help, self-care and self-medicate [32]. Therefore, it has been reported that some people resource to self-medication [25] and others to the use of medicinal plants [33] as potential but unproven methods to ameliorate and/or prevent symptoms related to COVID-19. The Ministry of Health in Peru published the symptomatic pharmacological treatment options for

COVID-19 to control the pandemic [34]. These options included the clinical use of hydroxy-chloroquine and ivermectin for mild cases of COVID-19 and hydroxychloroquine plus azi-thromycin and/or chloroquine phosphate plus ivermectin for moderate and severe cases of COVID-19 [7, 34, 35]. However, this recommendation led to self-medication reports in Peru [7, 25], which also gets accompanied with the use of medicinal plants. This practice has become increasingly common in Peru as panic spread in the general population, who anx-iously wait for positive news about the prevention, treatment and vaccination [25]. It has been reported that medicinal plants and bioactive compounds that previously exhibited antiviral activity against SARS-CoV-1 and MERS-CoV, could also exhibit potential efficacy against SARS-CoV-2 [36, 37]. This potential activity has been proposed to be due to their activity on the ACE-2 receptor, 3CLpro and other SARS-CoV-2 viral protein targets [36]. Another approach has been centered around the computational approaches to search for potential inhibitory compounds against the active binding pockets of SARS-CoV-2 target proteins [38]. It has been also reported that mesenchymal stem cells (MSCs) and their exosomes (MSCs-Exo) can significantly lower the risk of lung injuries related to alveolar inflammation and related pathological conditions, such as the one observed in COVID-19 patients [39].

The idiosyncrasy of the population has also focused on the use of medicinal plants, natural products or preparations, with antiviral and anti-inflammatory properties to strengthen the immune system or treat respiratory diseases [40]. Countries, such as China, India, Bolivia, Morocco, Nepal, Peru and Brazil, are using traditional medicine against COVID-19 [40–49]. There are approximately 50,000 plant species with medicinal uses, and the WHO has estimated that 80% of the population of developing countries use traditional medicine as their main source of medicines [50]. In Europe, the clinical use of medicinal plants is approved under the directive 24/2004 for the treatment of colds [51]; these treatments are accessible and available. In Latin America, the Regional Office of WHO for the Americas (AMOR/PAHO) indicates that 71% and 40% of the population of Chile and Colombia, respectively, use traditional medi-cine [52, 53]. In Peru, a megadiverse country, the use of medicinal plants for the treatment of various conditions, such as malaise and gastrointestinal and respiratory diseases, dates from the Inca period [41, 53–56], and currently approximately 1,400 species are used for medicinal purposes in both native and urban communities [41, 53]. It comes as no surprise that tradi-tional medicine is currently being used by the Peruvian population in the context of the COVID-19 pandemic. Thus, the objective of this study was to assess factors associated with the use of medicinal plants as an adjuvant for the treatment or prevention of respiratory symptoms during the COVID-19 pandemic in Cusco, Peru.

## Materials and methods

### Ethics statement

The survey was approved by the San Antonio Abad del Cusco National University ethics com-mittee (#007-2020-CBI-UNSAAC). Written consent was obtained from the participants before starting the survey. The participants remained anonymous and had the option to finish the survey at any time, and their information was kept confidential. All the survey participants were well-versed on the study intentions and were required to consent before the enrollment. The participants were not involved in any of the planning, execution and reporting stages of the study.

### Study design

We conducted an online cross-sectional multicenter survey, which was initially evaluated by 10 expert judges using Aiken's V [57]. After including the experts' observations, a pilot study

was performed (from August 16 to 24, 2020) with 336 respondents in in five districts of Cusco, Peru. The pilot data was used to calculate the minimal sample size necessary for the actual study. It was determined that a minimum sample size of 1,530 was necessary to achieve a minimum percentage difference of 2.5% (49.0% versus 51.5%), a statistical power of 80%, and a confidence level of 95%. The sample size was calculated using power analysis [58].

The actual survey consisted of an online questionnaire that was sent via WhatsApp, Messenger, and Facebook. The shared questionnaire was made anonymous ensuring data confidentiality and reliability. The survey was performed from August 31 to September 20, 2020 after approximately 9 months of lockdown and social distancing measures in Peru due to the COVID-19 outbreak. At the beginning of the survey (August 31) the number of COVID-19 confirmed cases was 652,037 and 28,944 deaths [59], while at the end of the survey (September 20) the confirmed cases increased to 772,896 and the deaths increased to 31,474 [60]. We surveyed general public who were adults of both genders aged 20 to 70 years in five districts of Cusco, Peru with high-risk COVID-19 transmission according to the Epidemiological Alert AE-017-2020 [61]. The five districts were Cusco, San Jerónimo, San Sebastián, Santiago, and Wanchaq. Participants were recruited by the research team of the Universidad Nacional de San Antonio Abad del Cusco. There were no exclusions because we implemented that it was mandatory to reply all the answers. Therefore, we only obtained complete and high-quality answers, which was verified by a data quality check.

## Outcomes and covariates

The survey (S1 Annex) included 11 questions, 5 were demographic questions, 2 were related to the use of medicinal plants as preventive or treatment of respiratory symptoms related to COVID-19, 2 were related to the diagnosis of COVID-19 in themselves and the close environment (family and friends), and the last 2 questions related to the medicinal plants were used and to what respiratory symptom(s) they were used for. The demographic questions included sex, age, education level, occupation or professional activity and the district of residence in Cusco.

The respondents were asked to indicate if during the COVID-19 lockdown they used medicinal plants to prevent or treat respiratory symptoms related to COVID-19. Then, the respondents were asked if they were diagnosed with COVID-19, and if any family member or friend was diagnosed with COVID-19. The respondents were asked to select from a list of 17 selected medicinal plants the ones they have used to prevent or treat COVID-19 related respiratory symptoms. The selection of the medicinal plants was based on ethnopharmacological reports of the ones used for respiratory problems in Peru [62–68]. Finally, the respondents were asked to select the symptom(s) why they consumed any of the medicinal plants indicated on the previous question. The symptoms included cough, sore throat, fever, headache or malaise. Loss of taste or smell, nausea/vomiting and diarrhea were not included. The selected symptoms relate to the most common COVID-19 symptoms reported by the Center of Disease Control and Prevention (CDC) [69].

## Statistical analysis

Data analysis was done in STATA version 14 (Stata Corp) with a significance level set at p<0.05. Descriptive analysis of categorical (demographic) variables was performed to show the frequency and percentage of each response. The results were summarized in unidimensional tables to identify the medicinal plants that are most used by the respondents. Chi-square test was performed in the bivariate analysis to determine the association between the studied variables. Generalized linear models were used in the multivariate analysis using the Poisson

family, the log-link function, and the models for variances of robust models and the district of residence as a cluster, thereby obtaining the prevalence ratios (PR). The 95% confidence intervals (CI) and p-value < 0.05 were considered as the limit of statistical significance.

## Results

### Sociodemographic characteristics of the respondents

A total of 1,747 respondents participated in this study. The majority of the study participants were female [59.1% (1,033)], the median age was 31 years (interquartile range: 24–41 years), 29.7% (518) were university and high school students, 59.2% (1,035) had higher education, and 33.3% (582) lived in the district of Cusco. Concerning to the COVID-19 questions and use of medicinal plants, 12.2% (214) had COVID-19, 65.9% (1,151) had a family member or friend with COVID-19, 80.2% (1,401) used medicinal plants to prevent respiratory symptoms, and 71.0% (1,241) used medicinal plants to treat respiratory symptoms (**Table 1**).

### Use of medicinal plants as preventive or treatment for respiratory symptoms

As shown on **Table 2**, the use of medicinal plants for the prevention of respiratory symptoms was associated with gender (p < 0.001), age (p < 0.001), occupation (p < 0.001), education (p < 0.001), and whether a family member or friend had COVID-19 (p < 0.001). However, no association was observed with the district of residence (p = 0.702) and if the respondent had COVID-19 (p = 0.632). In the case of the use of medicinal plants for the treatment of respiratory symptoms, the bivariate analysis showed that it was associated with gender (p = 0.013), age (p < 0.001), occupation (p < 0.001), education (p < 0.001), district of residence (p = 0.005), whether the respondent had COVID-19 (p < 0.001), and whether a family member or friend had COVID-19 (p < 0.001).

### Medicinal plants used and respiratory symptoms associated with their use as treatment options

As shown on **Table 3**, the respondents reported the use of medicinal plants from a preselected list for the treatment of COVID-19 related respiratory symptoms. The most frequently used medicinal plant was eucalyptus (*Eucalyptus globulus* Labill.) followed by ginger (*Zingiber officinale* Roscoe), garlic (*Allium sativum* L.), matico (*Piper aduncum* L.), chamomile (*Matricaria recutita* L.) and coca (*Erythroxylum coca* Lam.). It was observed that all the medicinal plants were used for 2 or more respiratory symptoms. The bivariate statistics showed that the use of medicinal plants was associated with the occurrence of two or more symptoms (24–51%), followed by malaise (11–41%). In addition, there was a difference in consumption according to the type of symptom (p < 0.041) for all medicinal plants except panty (*Cosmos peucedanifolius* Wedd.) (p = 0.076).

### Multivariate analysis of the factors associated to the use of medicinal plants

In the multivariate analysis (**Table 4**) of the use of medicinal plants for the treatment or prevention of respiratory symptoms during the COVID-19 pandemic found a positive association with the use of eucalyptus (*Eucalyptus globulus* Labill.) for treatment (PR: 1.26, 95% CI: 1.16–1.37, p < 0.001) and prevention (PR: 1.24, 95% CI: 1.15–1.35, p < 0.001). Followed by the use of matico (*Piper aduncum* L.) for treatment (PR: 1.20, 95% CI: 1.07–1.34, p = 0.011) and prevention (PR: 1.12, 95% CI: 1.06–1.19, p < 0.001). In addition, there is also a positive association between the use of ginger for treatment (PR: 1.13, 95% CI: 1.03–1.25, p = 0.011) and

**Table 1. Socio-demographic characteristics of respondents that used medicinal plants for the treatment or prevention of respiratory symptoms during the COVID-19 pandemic in Cusco, Peru.**

| Variable | N | % |
|---|---|---|
| **Gender** | | |
| Female | 1033 | 59.1% |
| Male | 714 | 40.9% |
| **Age (years)[a]** | 31 | 24–41 |
| **Occupation or professional activity** | | |
| Housewife | 195 | 11.1% |
| Self-employed | 386 | 22.1% |
| Public sector | 323 | 18.5% |
| Private sector | 243 | 13.9% |
| Student | 518 | 29.7% |
| Other | 82 | 4.7% |
| **Education** | | |
| No education | 33 | 1.9% |
| Primary | 58 | 3.3% |
| Secondary | 328 | 18.8% |
| Technical | 293 | 16.8% |
| University | 1035 | 59.2% |
| **District of residence** | | |
| Cusco | 582 | 33.3% |
| San Jerónimo | 272 | 15.6% |
| San Sebastián | 332 | 19.0% |
| Santiago | 278 | 15.9% |
| Wanchaq | 283 | 16.2% |
| **Diagnosed with COVID-19** | | |
| No | 1533 | 87.8% |
| Yes | 214 | 12.2% |
| **Family member or friend diagnosed with COVID-19** | | |
| No | 596 | 34.1% |
| Yes | 1151 | 65.9% |
| **Prevention for respiratory symptoms** | | |
| Did not use plants for prevention | 346 | 19.8% |
| Used plants for prevention | 1401 | 80.2% |
| **Treatment of respiratory symptoms** | | |
| Did not use plants for treatment | 506 | 29.0% |
| Used plants for treatment | 1241 | 71.0% |

[a]Median and interquartile range.

prevention (PR: 1.08, 95% CI: 1.01–1.16, p = 0.021), the use of garlic (*Allium sativum* L.) for prevention only (PR: 1.06, 95% CI: 1.01–1.11, p = 0.023), and the use of chamomile (*Matricaria recutita* L.) for treatment only (PR: 1.12, 95% CI: 1.03–1.23, p = 0.011).

There was also a negative association between the use of wira wira (*Ganaphalium viravira* Molina) for treatment (PR: 0.9, 95% CI: 0.82–0.98, p = 0.016) and prevention (PR: 0.89, 95% CI: 0.85–0.93, p < 0.001) and the use of panty for treatment only (PR: 0.87, 95% CI: 0.78–0.97, p = 0.009). Therefore, eucalyptus (*Eucalyptus globulus* Labill.), matico (*Piper aduncum* L.), ginger (*Zingiber officinale* Roscoe), and chamomile (*Matricaria recutita* L.) were the most used

**Table 2. Use of medicinal plants as preventive or treatment for respiratory symptoms during the COVID-19 pandemic in Cusco, Peru.**

| Variable | Used medicinal plant as preventive | | p-value | Used medicinal plant as treatment | | p-value |
|---|---|---|---|---|---|---|
| | No | Yes | | No | Yes | |
| **Gender** | | | | | | |
| Female | 175 (16.9%) | 858 (83.1%) | <0.001 | 276 (26.7%) | 757 (73.3%) | 0.013 |
| Male | 171 (24.0%) | 543 (76.0%) | | 230 (32.2%) | 484 (67.8%) | |
| **Age** (years)* | 28 (23–39) | 31 (24–42) | <0.001 | 28 (23–40) | 32 (25–42) | <0.001 |
| **Occupation or professional activity** | | | | | | |
| Housewife | 18 (9.2%) | 177 (90.8%) | <0.001 | 34 (17.4%) | 161 (82.6%) | <0.001 |
| Self-employed | 70 (18.1%) | 316 (81.9%) | | 93 (24.1%) | 293 (75.9%) | |
| Public sector | 58 (18.0%) | 265 (82.0%) | | 82 (25.4%) | 241 (74.6%) | |
| Private sector | 38 (15.6%) | 205 (84.4%) | | 63 (25.9%) | 180 (74.1%) | |
| Student | 152 (29.3%) | 366 (70.7%) | | 210 (40.5%) | 308 (59.5%) | |
| Other | 10 (12.2%) | 72 (87.8%) | | 24 (29.3%) | 58 (70.7%) | |
| **Education** | | | | | | |
| No education | 10 (30.3%) | 23 (69.7%) | <0.001 | 10 (30.3%) | 23 (69.7%) | <0.001 |
| Primary | 7 (12.1%) | 51 (87.9%) | | 11 (19.0%) | 47 (81.0%) | |
| Secondary | 42 (12.8%) | 286 (87.2%) | | 61 (18.6%) | 267 (81,4%) | |
| Technical | 46 (15.7%) | 247 (84.3%) | | 72 (24.6%) | 221 (75.4%) | |
| University | 241 (23.3%) | 794 (76.7%) | | 352 (34.0%) | 683 (66.0%) | |
| **District of residence** | | | | | | |
| Cusco | 122 (21.0%) | 460 (79.0%) | 0.702 | 200 (34.4%) | 382 (65.6%) | 0.005 |
| San Jerónimo | 59 (21.7%) | 213 (78.3%) | | 76 (28.0%) | 196 (72.0%) | |
| San Sebastián | 63 (19.0%) | 269 (81.0%) | | 81 (24.4%) | 251 (75.6%) | |
| Santiago | 51 (18.4%) | 227 (81.6%) | | 67 (24.1%) | 211 (75.9%) | |
| Wanchaq | 51 (18.0%) | 232 (82.0%) | | 82 (29.0%) | 201 (71.0%) | |
| **Had COVID-19** | | | | | | |
| No | 301 (19.6%) | 1232 (80.4%) | 0.632 | 474 (30.9%) | 1059 (69.1%) | <0.001 |
| Yes | 45 (21.0%) | 169 (79.0%) | | 32 (15.0%) | 182 (85.0%) | |
| **Family member or friend diagnosed with COVID-19** | | | | | | |
| No | 158 (26.5%) | 438 (73.5%) | <0.001 | 225 (37.8%) | 371 (62.2%) | <0.001 |
| Yes | 188 (16.3%) | 963 (83.7%) | | 281 (24.4%) | 870 (75.6%) | |

The p-values were obtained with chi-square tests and the sum of ranges (for age). The descriptive values for age are the median (interquartile ranges).

for the treatment of respiratory symptoms, whereas panty (*Cosmos peucedanifolius* Wedd.) and wira wira (*Ganaphalium viravira* Molina) were the least used. As for prevention, eucalyptus (*Eucalyptus globulus* Labill.), matico (*Piper aduncum* L.), ginger (*Zingiber officinale* Roscoe), and garlic (*Allium sativum* L.) were the most used, whereas wira wira (*Ganaphalium viravira* Molina) was the least used.

Regarding the adjustment of demographic variables and the use of medicinal plants for the treatment or prevention of respiratory symptoms, it is important to mention that there was a positive association between age and prevention (PR: 1.00, 95% CI: 1.00–1.01, p = 0.046). Precisely, older respondents used more medicinal plants for prevention. There was also a positive association between the respondents diagnosed with COVID-19 and the use of medicinal plants for treatment (PR: 1.22, 95% CI: 1.11–1.34, p < 0.001); precisely, those with COVID-19 used more medicinal plants for prevention. There was also a positive association between the respondents with a family member or friend diagnosed with COVID-19 and the use of medicinal plants for treatment (PR: 1.11, 95% CI: 1.06–1.17, p < 0.001) and prevention (PR: 1.08,

**Table 3. Percentage of the use of medicinal plants for the treatment of respiratory symptoms during the COVID-19 pandemic in Cusco, Peru.**

| Medicinal plant | | N | Two or more symptoms | Cough | Sore throat | Malaise | Fever | Headache | Other symptoms | p-value |
|---|---|---|---|---|---|---|---|---|---|---|
| Common name | Scientific name | | | | | | | | | |
| Eucalyptus | *Eucalyptus globulus* Labill. | 989 | 48% | 20% | 6% | 17% | 1% | 1% | 7% | <0,001 |
| Ginger | *Zingiber officinale* Roscoe | 927 | 46% | 17% | 20% | 11% | 1% | 0% | 5% | 0,001 |
| Garlic | *Allium sativum* L. | 838 | 46% | 21% | 16% | 11% | 0% | 0% | 6% | <0,001 |
| Matico | *Piper aduncum* L. | 661 | 50% | 13% | 10% | 17% | 1% | 1% | 8% | 0,001 |
| Chamomile | *Matricaria recutita* L. | 642 | 38% | 3% | 8% | 32% | 2% | 6% | 11% | <0,001 |
| Coca | *Erythroxylum coca* Lam. | 474 | 42% | 7% | 14% | 23% | 1% | 4% | 9% | 0,040 |
| Muña | *Minthostachys acris* Schmidt-Leb. | 451 | 32% | 10% | 6% | 32% | 1% | 3% | 16% | <0,001 |
| Oregano | *Origanum vulgare* L. | 346 | 26% | 11% | 7% | 37% | 1% | 1% | 16% | <0,001 |
| Rosemary | *Rosmarinus officinalis* L. | 298 | 24% | 8% | 5% | 41% | 3% | 4% | 15% | <0,001 |
| Panty | *Cosmos peucedanifolius* Wedd. | 211 | 41% | 30% | 6% | 15% | 1% | 0% | 7% | 0,076 |
| Lemon balm | *Melissa officinalis* L. | 166 | 27% | 7% | 6% | 31% | 1% | 4% | 24% | 0,013 |
| Thyme | *Thymus vulgaris* L. | 164 | 35% | 9% | 5% | 36% | 2% | 0% | 12% | <0,001 |
| Sage | *Salvia officinalis* L. | 146 | 42% | 15% | 4% | 21% | 3% | 1% | 13% | <0,001 |
| Keto-keto | *Gnaphalium coarctacum* Willd. | 126 | 40% | 25% | 6% | 13% | 4% | 2% | 10% | 0,010 |
| Geranium | *Geranium sibiricum* L. | 106 | 26% | 12% | 10% | 29% | 5% | 1% | 17% | 0,004 |
| Asmachilca | *Aristeguietia gayana* (Wedd.) | 102 | 51% | 10% | 4% | 17% | 1% | 1% | 16% | 0,014 |
| Wira wira | *Ganaphalium viravira* Molina | 83 | 40% | 15% | 6% | 19% | 2% | 1% | 17% | 0,014 |

The p-values were obtained based on the chi-square test.

95% CI: 1.04–1.13, p < 0.001), but the respondents used fewer plants for treatment if they had a technical or higher education (PR: 0.89, 95% CI: 0.83–0.95, p < 0.001).

## Discussion

Regarding the prevention of COVID-19 respiratory symptoms, this study showed that 80.2% of the population of Cusco, Peru, used medicinal plants for this purpose. Comparatively, a study conducted in a population of the state of Querétaro in Mexico showed that the main conditions treated with medicinal plants were asthma (18.42%), bronchitis (2.6%), flu (5.2%), congestion in the respiratory tract (10.5%), sore throat (21%), throat infection (15.7%), pneumonia (5.2%), sinusitis (10.55%), cough (55.2%), and tuberculosis (2.6%) [70]. Additionally, there are populations in many regions of the world that are using medicinal plants for the prevention of COVID-19, because these plants are more readily available than Western medicine. In this regard, a study performed in the Moroccan population has mentioned medicinal plants similar to those reported in this study, such as eucalyptus, garlic, onion, ginger, thyme, turmeric, and rosemary [71].

Regarding the use of medicinal plants for the treatment of COVID-19 respiratory symptoms, 29.0% of the respondents did not use any plant, whereas 71.0% did. One of the most relevant clinical manifestations of COVID-19 is the great damage on the respiratory tract, causing respiratory distress that can lead to death, and for this reason, effective and non-invasive treatments are required [72]. This includes the use of medicinal plants, which were revalued during this pandemic to manage the COVID-19 symptoms, because plants are a source of plant metabolites with antiviral activity [73]. In this context, a study carried out in Bolivia evaluated eucalyptus, wira wira, and chamomile for their antibacterial, anti-inflammatory, and fungicidal properties [40]. Alternatively, it has been reported that diet supplementation with probiotics and nutraceuticals play a fundamental role in the treatment of respiratory symptoms,

**Table 4. Multivariate analysis of the use of medicinal plants for the treatment or prevention of respiratory symptoms during the COVID-19 pandemic in Cusco, Peru.**

| Medicinal Plant | | For treatment | For prevention |
|---|---|---|---|
| Common Name | Scientific Name | | |
| Eucalyptus | *Eucalyptus globulus* Labill. | 1.26 (1.16–1.37) p<0.001 | 1.24 (1.15–1.35) p<0.001 |
| Ginger | *Zingiber officinale* Roscoe | 1.13 (1.03–1.25) p = 0.011 | 1.08 (1.01–1.16) p = 0.021 |
| Garlic | *Allium sativum* L. | p = 0.121 | 1.06 (1.01–1.11) p = 0.023 |
| Coca | *Erythroxylum coca* Lam. | p = 0.517 | p = 0.596 |
| Muña | *Minthostachys acris* Schmidt-Leb. | p = 0.207 | p = 0.263 |
| Matico | *Piper aduncum* L. | 1.20 (1.07–1.34) p = 0.002 | 1.12 (1.06–1.19) p<0.001 |
| Chamomile | *Matricaria recutita* L. | 1,12 (1.03–1.23) p = 0.011 | p = 0.0151 |
| Rosemary | *Rosmarinus officinalis* L. | p = 0.167 | p = 0.211 |
| Oregano | *Origanum vulgare* L. | p = 0.126 | p = 0.817 |
| Lemon balm | *Melissa officinalis* L. | p = 0.130 | p = 0.697 |
| Geranium | Geranium, L. | p = 0.526 | p = 0.428 |
| Thyme | *Thymus vulgaris* L. | p = 0.157 | p = 0.063 |
| Panty | *Cosmos peucedanifolius* Wedd. | 0.87 (0.78–0.97) p = 0.009 | p = 0.908 |
| Keto-keto | *Gnaphalium coarctacum* Willd. | p = 0.080 | p = 0.262 |
| Sage | *Salvia officinalis* L. | p = 0.0519 | p = 0.431 |
| Wira Wira | *Ganaphalium viravira* Molina | 0.90 (0.82–0.98) p = 0.016 | 0.89 (0.85–0.93) p<0.001 |
| Asmachilca | *Aristeguietia gayana* (Wedd.) | p = 0.742 | p = 0.466 |
| **Adjustment variables** | | | |
| Age (Years)* | | p = 0.054 | 1.00 (1.00–1.01) p = 0.046 |
| Male | | p = 0.708 | p = 0.105 |
| Technical or higher education | | 0.89 (0.83–0.95) p<0.001 | p = 0.106 |
| Had COVID-19 diagnosis | | 1.22 (1.11–1.34) p<0.001 | p = 0.472 |
| Family member or friend diagnosed with COVID-19 | | 1.11 (1.06–1.17) p<0.001 | 1.08 (1.04–1.13) p<0.001 |

*The variable age was considered quantitatively.

because many products produce an immune response to respiratory viruses in addition to their regulatory activity for the inflammation caused by COVID-19 [74]. The therapeutic use of medicinal plants has increased in many Latin American countries over time [75]. Our study reported that 80% of the respondents used medicinal plants when they or their family member or friend had COVID-19, which correlates to previous reports [76].

As for the socio-educational factors associated with the use of medicinal plants for the prevention of respiratory symptoms, our study reported that female respondents (83.1%) used them. This correlates to previous studies where women are more versed in the properties of medicinal plants [77] and that they typically use medicinal plants to take care of the health of their family members [78, 79]. Therefore, probably in most populations, women are those who transmit the traditional domestic knowledge from generation to generation [80]. Sighal obtained different results in a study performed in 2005 on the role of gender in the use and management of medicinal plants in indigenous communities of India, as it was found that both females and males had the knowledge and appreciation of their use [81]. The study of Biniam et al. found a greater tendency for women to use medicinal plants than men [82]. This relevant role of women is not only in medicinal plants but also in food safety practices [83]. This factor is relevant in the current context of the COVID-19 pandemic, and it is relevant to focus on females as an important element for the prevention and rational treatment of patients with COVID-19.

Regarding the significant association between the use of medicinal plants with primary and secondary education, this is explained by the fact that those with professional training are more likely to use a scientifically validated treatment, abandoning the use of medicinal plants. It has been reported that elder and low education people could be at higher risk of disease complications because they prefer the use of medicinal plants over the adherence to pharmacological treatment [84]. This could potentially be riskier for patients with COVID-19.

For the treatment of COVID-19 respiratory symptoms, 24 to 51% of respondents in our study used medicinal plants when there were more than two symptoms and 11 to 41% when they presented malaise. There are studies describing the ethnomedicinal use during the COVID-19 pandemic of different communities and cultures around the world, especially in Asian countries, such as India, China, Japan, and Pakistan and some parts of Africa [85]. COVID-19 symptoms develop with inflammation and hemotoxicity, which could suggest that blood-purifying plants with anti-inflammatory, antioxidant, and antiviral properties could be considered as candidates for the treatment of COVID-19 [86]. There are also herbal remedies, such as those made from *Uncaria tomentosa* or cat's claw, a climbing vine that grows in the Peruvian jungle, which is used to stimulate the immune system [85]. The most used plants in our study included eucalyptus (*Eucalyptus globulus* Labill.), garlic (*Allium sativum* L.), lemon balm (*Melissa officinalis* L.), and geranium (*Geranium sibiricum* L.). It has been reported that eucalyptus (*Eucalyptus globulus* Labill.) is an effective antiviral agent against SARS-CoV-2 for its eucalyptol content, which was assessed in molecular docking studies [85]. Moreover, it has been reported that jensenone, a compound obtained from the essential oil of eucalyptus exhibits antiviral effect against the main protein of SARS-CoV-2 [85]. Additionally, garlic (*Allium sativum* L.) exhibited an inhibitory effect on SARS-CoV-2 replication; thus, it is a promising agent against COVID-19 [87]. A similar effect was found for palillo, a curcumin extracted from turmeric [87], as determined by molecular docking studies [85]. A meta-analysis study performed on medicinal plants suggested that plants, such as turmeric, can be used as a prophylaxis against SARS-CoV-2 according to docking studies that suggest its use; therefore, more trials should be carried out [86]. Another study was performed on natural molecules from plants with antiviral properties such as rosemary and cinnamon, reporting that they present low toxicity, and abundant active ingredients that can be used against viral infections [88]. There are also other studies on medicinal plants, such as ginger (*Zingiber officinale* Roscoe), whose rhizome has been used to alleviate fever and other COVID-19 symptoms in Africa [89]. The essential oil of eucalyptus (*Eucalyptus globulus* Labill.) has been reported to enhance the innate cell-mediated immune response and can be used in infectious diseases as an immunoregulatory agent [90]. An analysis of the essential oil detected 11 bioactive compounds such as 1.8 cineole (85.8%), α-pinene (7.2%) and β- myrcene (1.5%) [91]. Other compounds identified in the oil were β-pinene, limonene, α-phelandrene, γ-terpinene, linalol, pinocarveol, terpinen-4-ol and α-terpineol, that exhibited antimicrobial effects [91].

In the case of ginger (*Zingiber officinale* Roscoe), a randomized controlled study was performed to assess its effects on respiratory manifestations in patients with acute respiratory syndrome due to COVID-19 [92]. The experimental group was administered the standard treatment for COVID-19 according to the protocol of the Iranian Ministry of Health along with ginger tablets (Vomigone®) in a dose of 1000 mg, 3 times a day, for a period of seven days [92]. An improvement in clinical symptoms was evidenced within 7 days of treatment including fever, dry cough, fatigue and other symptoms such as thrombocytopenia, lymphocytopenia and C-reactive protein [92]. The consumption of ginger has been attributed to have properties against pneumonia and pulmonary fibrosis, and in the latter case it reduces oxidative stress and the inflammatory response in animal models that were chemically induced with pulmonary fibrosis [93].

Garlic (*Allium sativum* L.) is consumed around the world as a condiment and is an important part of traditional Chinese and Indian medicine since its active principles are organosulfides, saponins and polysaccharides [94]. Its immunomodulatory activity is mainly due to the polysaccharides as they regulate the homeostasis of the immune system, maintain the immune response and the expression and proliferation of cytokine genes [94]. The bioactive compounds present in garlic have potential effects on respiratory tract infections, intra-alveolar edema, pulmonary fibrosis, sepsis, and acute lung injury [93]. Its active principles: allicin, s-allyl cysteine (SAC), alliin and diallyl thiosulfonate (allicin) showed antiviral, antifibrotic, antioxidant, anti-inflammatory and immunomodulatory properties [93]. Asmachilca (*Aristeguietia gayana* (Wedd.)) is used for its expectorant effect and in asthma cases [95]. However, caution needs to be applied because it contains 1,2-dehydropyrrolizidine alkaloid esters [96], which have been reported to potentially cause hepatotoxicity, pneumotoxicity, genotoxicity and carcinogenicity [97]. It is important to mention that certain medicinal plants reported as potential complementarity treatments for COVID-19 can contain compounds that could be harmful [37]. For instance, *Echinacea purpurea* can increase the release of IL-1, IL-10 and TNF-α by macrophages [98, 99]. Thus, causing hypercytokinaemia or the increase of proinflammatory cytokines that can cause complication in COVID-19 patients [37, 100]. Another example is *Chinchona sp.* because it contains quinine, which has a mode of action similar to chloroquine [101]. Quinine has been reported to have a dual effect related to immune response, it acts as immunostimulator when it effectively intensifies the production of IFN-α [37]. However, it can also inhibit the release of TNF-α causing an immunosuppressant effect [37]. This dual effect should caution healthy people to constantly consume *Chinchona sp.* as a COVID-19 preventive because of the potential harmful effect it can cause [37].

Peru is one of the countries that have a wide pantry of medicinal plants that are one of the main alternatives in health care to prevent and treat various diseases. In different countries, there is also a worldwide wealth of knowledge, theories, and practices on the use of plants as natural medicines for the treatment of diseases. Medicinal plants have been used since prehistoric times–a tradition that has been passed down from generation to generation. Traditionally, medicinal plants are consumed as the fresh form (i.e. ginger) or dry leaves, both in infusions with hot water. The World Health Organization (WHO) considers the Natural and Traditional Medicine, which includes treatment with medicinal plants, as the most natural, safe, effective, and affordable medicine [102]. The use of medicinal plants for respiratory conditions has also been reported in various parts of the world from China [46], India [44], Saudi Arabia [103] to Mexico [104] and Ecuador [105]. However, it needs to be acknowledged that the ethnopharmacological use of medicinal plants for prevention or treatment of respiratory symptoms related to COVID-19 still needs to be evaluated in clinical settings in order to have solid evidence of their effectiveness and to isolate compounds with potential pharmacological use. Another important factor to evaluate in more detail is the effect that the COVID-19 pandemic in the dynamics of the community as well as the SARS-CoV-2 prevalence and fate in environmental matrices, which could help policy maker to develop mitigation strategies [26, 106, 107].

The limitations of this study included the fact that the results cannot be extrapolated to the entire Peruvian population. The objective of this study was to determine the association between the use of medicinal plants and the treatment or prevention of respiratory symptoms in the population of the five districts of Cusco, one of the most important cities in Peru. However, this study is the first to investigate this association in this population; therefore, this could become a basis for other studies that could cover a larger population from all over the country.

Another limitation was the selection bias cause by not having performed random sampling to obtain the responses. Because of the nature of the study (cross-sectional study design) we could not determine definitive cause and effect associations. Similarly, the responders performed a self-reported assessment in an online data collection platform, which could lead to under or over-reporting and the data collector has not ability to verify or validate. Another limitation was that we did not assess frequency of consumption of medicinal plants, nor the amount of plant consumed.

## Conclusions

The current study reported an association between the use of 17 medicinal plants and the treatment or prevention of the respiratory symptoms related to COVID-19, and the most used plants were eucalyptus, ginger, spiked pepper, chamomile, and garlic. Moreover, it was determined that the study population used a greater number of plants for disease prevention when the respondent was older and if they or a friend or family member had contracted COVID-19. It was also observed that respondents with technical or higher education used less plants for treatment. The potential use of medicinal plants for respiratory conditions is acknowledged but more research is necessary to have solid evidence of their effectiveness and to isolate compounds with potential pharmacological use. Further studies are warranted to determine proper doses, forms of preparation and potential combination of these medicinal plants.

## Supporting information

**S1 Annex. Survey to assess the use of medicinal plants in the prevention and treatment of respiratory symptoms during the COVID-19 pandemic.**
(DOCX)

**S2 Annex. Survey to assess the use of medicinal plants in the prevention and treatment of respiratory symptoms during the COVID-19 pandemic in Spanish, the original language.**
(DOCX)

## Acknowledgments

We thank all the participants in the study.

## Author Contributions

**Conceptualization:** Magaly Villena-Tejada, Christian R. Mejia, Jaime A. Yañez.

**Data curation:** Magaly Villena-Tejada, Ingrid Vera-Ferchau, Anahí Cardona-Rivero, Rina Zamalloa-Cornejo, Maritza Quispe-Florez, Zany Frisancho-Triveño, Susan G. Alvarez-Sucari.

**Formal analysis:** Magaly Villena-Tejada, Ingrid Vera-Ferchau, Anahí Cardona-Rivero, Rina Zamalloa-Cornejo, Zany Frisancho-Triveño, Rosario C. Abarca-Meléndez, Susan G. Alvarez-Sucari, Christian R. Mejia.

**Funding acquisition:** Magaly Villena-Tejada.

**Investigation:** Magaly Villena-Tejada, Ingrid Vera-Ferchau, Maritza Quispe-Florez, Rosario C. Abarca-Meléndez, Christian R. Mejia.

**Methodology:** Magaly Villena-Tejada, Ingrid Vera-Ferchau, Anahí Cardona-Rivero, Rina Zamalloa-Cornejo, Zany Frisancho-Triveño, Rosario C. Abarca-Meléndez, Susan G. Alvarez-Sucari, Christian R. Mejia.

**Project administration:** Magaly Villena-Tejada, Jaime A. Yañez.

**Resources:** Magaly Villena-Tejada.

**Software:** Rina Zamalloa-Cornejo, Zany Frisancho-Triveño, Christian R. Mejia.

**Supervision:** Magaly Villena-Tejada, Christian R. Mejia, Jaime A. Yañez.

**Validation:** Rina Zamalloa-Cornejo, Christian R. Mejia, Jaime A. Yañez.

**Visualization:** Rina Zamalloa-Cornejo.

**Writing – original draft:** Magaly Villena-Tejada, Maritza Quispe-Florez, Rosario C. Abarca-Meléndez, Susan G. Alvarez-Sucari, Christian R. Mejia, Jaime A. Yañez.

**Writing – review & editing:** Christian R. Mejia, Jaime A. Yañez.

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
