## [Decision Letter · Decision Letter 0]

14 Jun 2021

PONE-D-21-15800

Use of medicinal plants for COVID-19 prevention and respiratory symptom treatment during the pandemic in Cusco, Peru: A cross-sectional survey

PLOS ONE

Dear Dr. Yanez,

Thank you for submitting your manuscript to PLOS ONE. After careful consideration, we feel that it has merit but does not fully meet PLOS ONE’s publication criteria as it currently stands. Therefore, we invite you to submit a revised version of the manuscript that addresses the points raised during the review process.

We look forward to receiving your revised manuscript.

Kind regards,

Mohd Adnan, PhD

Academic Editor

PLOS ONE

Additional Editor Comments:

Reviewers have raised some serious concerns and shortcomings in the study. MAJOR revision is suggested, and requires substantial and thorough revision to appreciate the quality of the manuscript. Therefore, authors are requested to revise their manuscript in light of reviewer's comments. Please justify and discuss all questions raised by the reviewers.

Journal Requirements:

"The authors declare that they have no competing interests."

We note that one or more of the authors are employed by a commercial company: Teoma Global.

4.1. Please provide an amended Funding Statement declaring this commercial affiliation, as well as a statement regarding the Role of Funders in your study. If the funding organization did not play a role in the study design, data collection and analysis, decision to publish, or preparation of the manuscript and only provided financial support in the form of authors' salaries and/or research materials, please review your statements relating to the author contributions, and ensure you have specifically and accurately indicated the role(s) that these authors had in your study. You can update author roles in the Author Contributions section of the online submission form.

4.2. Please also provide an updated Competing Interests Statement declaring this commercial affiliation along with any other relevant declarations relating to employment, consultancy, patents, products in development, or marketed products, etc.  

Reviewers' comments:

Reviewer's Responses to Questions

**Comments to the Author**

1. Is the manuscript technically sound, and do the data support the conclusions?

Reviewer #1: Yes

Reviewer #2: Partly

Reviewer #3: Yes

Reviewer #4: Partly

Reviewer #5: Partly

2. Has the statistical analysis been performed appropriately and rigorously? 

Reviewer #1: Yes

Reviewer #2: Yes

Reviewer #3: Yes

Reviewer #4: No

Reviewer #5: I Don't Know

3. Have the authors made all data underlying the findings in their manuscript fully available?

Reviewer #1: Yes

Reviewer #2: Yes

Reviewer #3: Yes

Reviewer #4: Yes

Reviewer #5: No

4. Is the manuscript presented in an intelligible fashion and written in standard English?

Reviewer #1: No

Reviewer #2: Yes

Reviewer #3: Yes

Reviewer #4: Yes

Reviewer #5: Yes

5. Review Comments to the Author

Reviewer #1: Although the manuscript titled: Use of medicinal plants for COVID-19 prevention and respiratory symptom treatment during the pandemic in Cusco, Peru: A cross-sectional survey. is interested and helpful for the people who looks for natural treatment to reduce the risk of COVID-19, but there are some points should be consider before publication:

1- Please be specific in your explanation and comparison of the doses used for each medicinal plant to prevent and treat COVID-19.

2- The authors did not specify which medicinal plant was effective or very effective in preventing or treating COVID-19; please be specific in your explanation.

3- The authors use a combination of medicinal plants to prevent or treat COVID-19; perhaps the combination has a greater effect than a single plant; please be specific in your explanation.

4- Please specify whether the authors used oral, injection, or smelling the steam of hot water from the medicinal plant for prevention or treatment.

5- The authors should write a good conclusion and be specific in their evaluation of medicinal plants.

6- Editing and correction of the manuscript in English.

Reviewer #2: The manuscript entitled " Use of medicinal plants for COVID-19 prevention and respiratory symptom treatment during the pandemic in Cusco, Peru: A cross-sectional survey". Title, abstract and overall rationale of work to some extent is good. However, there are still some minor concerns, which needs to be addressed minor revision.

1. The content of the manuscript is too short, specifically introduction and material methods section. Moreover, mechanism section is not properly elaborated, which is important for the reproducibility of the research.

3. Authors have mentioned very little about COVID-19 in introduction. Without the start of fundamentals about the subject in any manuscript doesn’t provide understanding to all types of readers. Therefore, it should be reader friendly with a proper flow. Authors can use below mentioned references, which will help them in adding this paragraph in introduction and discussion section and can be cited.

DOI: 10.1039/d0ra06379g

doi:10.1080/07391102.2020.1802345.

doi:10.3390/plants9091244.

doi:10.1155/2020/8835986.

doi: 10.1007/s11356-020-12165-1

dOI: 10.1007/s12035-021-02318-9

4. I would suggest the authors to enhance your theoretical discussion and arrives your debate or argument.

5. Conclusion section must be elaborated and I highly suggest author to write future prospective.

Reviewer #3: Dear authors.

Since the beginning of the COVID-19 epidemic, people around the world have been under constant stress. Currently, there is no established pharmacological strategy for the prevention and/or treatment of a new coronavirus infection. In this situation, we only have to wait for the vaccine and strengthen immunity to keep ourselves and our loved ones healthy. The topic touched upon in the article is relevant. The scientific content of the manuscript justifies its publication. The authors should justify the list of used medicinal plants in the questionnaire.

Reviewer #4: The manuscript with the title "Use of medicinal plants for COVID-19 prevention and respiratory symptom treatment during the pandemic in Cusco, Peru: A cross-sectional survey" describes and analyzed some data to know some factors associated with the use of medicinal plants as an adjuvant for Covid19 treatment or prevention.

Several analyses have shown the correlation of the medicinal plant consumption for the covid-19 and specific distinct population in Cusco, Peru. The authors claimed that the present article is the first investigation and could become a basis for other future studies.

The article is well written. However, several issues need to be resolved:

1. The authors should mention the reason why they select 17 medicinal plants in this study

2. The authors showed the respondent experience as Covid-19 patients (table 2); however, there was no data to confirm it. Please explain this condition

3. The data showed specific medicinal plants for the treatment Covid-19 (from each medicinal plant). Showing the combination of some medicinal plants for treating respiratory symptoms has become more interesting.

4. Several articles mention that some medicinal plants have a potential effect for Covid-19 treatment or prevention. However, the data of adverse effects from herbal medicine also cannot be excluded. The authors should raise this issue in the discussion. You can see this reference https://doi.org/10.1155/2020/2560645.

Reviewer #5: Dear authors

Please read my comments in the report attached here.

- Language and style should be checked and improved.

- Exact botanical names should be provided (www.theplantlist.org )

In Material and methods. What were the criteria used to include or exclude participants after the web-based investigation”?

- A diagram about the design of the study would add an important value to the paper. Please insert the number of participants in each step.

- How the authors calculated the sample? I need more details to judge the process.

- The selection seems to be biased. I need the exact process of participants’ selection.

- In the study protocol, participants were asked to select from a list of selected 17 medicinal species. This seems to be a serious bias. The authors should clarify and justify this approach.

- In the section results and discussion, replace the names of plants by their scientific names.

- For the mentioned species, important details are missing (parts used, administration routes, preparation, …..). This important information should be added.

Regards

---

## [Author Response · Author response to Decision Letter 0]

20 Aug 2021

Editor

We thank the editor for the positive review of our manuscript. The editor has made some critical and insightful comments that have definitely improved the final version. We have carefully amended the paper as suggested by the editor.

Comments 

Reviewers have raised some serious concerns and shortcomings in the study. MAJOR revision is suggested, and requires substantial and thorough revision to appreciate the quality of the manuscript. Therefore, authors are requested to revise their manuscript in light of reviewer's comments. Please justify and discuss all questions raised by the reviewers.

We thank the editor for this comment, we have modified the entire document to follow the PLOS ONE’s style requirements.

2. Please provide additional details regarding participant consent. In the ethics statement in the Methods and online submission information, please ensure that you have specified what type you obtained (for instance, written or verbal, and if verbal, how it was documented and witnessed). If your study included minors, state whether you obtained consent from parents or guardians. If the need for consent was waived by the ethics committee, please include this information. Once you have amended this/these statement(s) in the Methods section of the manuscript, please add the same text to the “Ethics Statement” field of the submission form (via “Edit Submission”). For additional information about PLOS ONE ethical requirements for human subjects research, please refer to http://journals.plos.org/plosone/s/submission-guidelines#loc-human-subjects-research.

We thank the editor for this comment, we have modified the Ethics statement to include a sentence related to the written consent obtained from the participants.

We thank the editor for this comment, the survey in English was included as Annex 1 in the original manuscript. We have included as Annex 2 the survey in Spanish, the original language.

4. Thank you for stating the following in the Competing Interests section: "The authors declare that they have no competing interests." We note that one or more of the authors are employed by a commercial company: Teoma Global.

4.1. Please provide an amended Funding Statement declaring this commercial affiliation, as well as a statement regarding the Role of Funders in your study. If the funding organization did not play a role in the study design, data collection and analysis, decision to publish, or preparation of the manuscript and only provided financial support in the form of authors' salaries and/or research materials, please review your statements relating to the author contributions, and ensure you have specifically and accurately indicated the role(s) that these authors had in your study. You can update author roles in the Author Contributions section of the online submission form. Please also include the following statement within your amended Funding Statement. “The funder provided support in the form of salaries for authors [insert relevant initials], but did not have any additional role in the study design, data collection and analysis, decision to publish, or preparation of the manuscript. The specific roles of these authors are articulated in the ‘author contributions’ section.” If your commercial affiliation did play a role in your study, please state and explain this role within your updated Funding Statement.

We thank the editor for this comment, we have included the mentioned details to the Funding and Competing Interests sections.

4.2. Please also provide an updated Competing Interests Statement declaring this commercial affiliation along with any other relevant declarations relating to employment, consultancy, patents, products in development, or marketed products, etc. Within your Competing Interests Statement, please confirm that this commercial affiliation does not alter your adherence to all PLOS ONE policies on sharing data and materials by including the following statement: "This does not alter our adherence to PLOS ONE policies on sharing data and materials.” (as detailed online in our guide for authors http://journals.plos.org/plosone/s/competing-interests) . If this adherence statement is not accurate and there are restrictions on sharing of data and/or materials, please state these. Please note that we cannot proceed with consideration of your article until this information has been declared. Please include both an updated Funding Statement and Competing Interests Statement in your cover letter. We will change the online submission form on your behalf. Please know it is PLOS ONE policy for corresponding authors to declare, on behalf of all authors, all potential competing interests for the purposes of transparency. PLOS defines a competing interest as anything that interferes with, or could reasonably be perceived as interfering with, the full and objective presentation, peer review, editorial decision-making, or publication of research or non-research articles submitted to one of the journals. Competing interests can be financial or non-financial, professional, or personal. Competing interests can arise in relationship to an organization or another person. Please follow this link to our website for more details on competing interests: http://journals.plos.org/plosone/s/competing-interests

We thank the editor for this comment, we have included the mentioned details to the Funding and Competing Interests sections.

5. We note that you have indicated that data from this study are available upon request. PLOS only allows data to be available upon request if there are legal or ethical restrictions on sharing data publicly. For information on unacceptable data access restrictions, please see http://journals.plos.org/plosone/s/data-availability#loc-unacceptable-data-access-restrictions. In your revised cover letter, please address the following prompts:

a. If there are ethical or legal restrictions on sharing a de-identified data set, please explain them in detail (e.g., data contain potentially identifying or sensitive patient information) and who has imposed them (e.g., an ethics committee). Please also provide contact information for a data access committee, ethics committee, or other institutional body to which data requests may be sent.

b. If there are no restrictions, please upload the minimal anonymized data set necessary to replicate your study findings as either Supporting Information files or to a stable, public repository and provide us with the relevant URLs, DOIs, or accession numbers. Please see http://www.bmj.com/content/340/bmj.c181.long for guidelines on how to de-identify and prepare clinical data for publication. For a list of acceptable repositories, please see http://journals.plos.org/plosone/s/data-availability#loc-recommended-repositories. We will update your Data Availability statement on your behalf to reflect the information you provide.

We thank the editor for this comment, there are no restrictions for the data. We have uploaded the anonymized data set at the Dryad data repository, and have modified the Supporting Information statement to: Anonymized data set supporting the findings of this study is stored at the Dryad data repository (https://datadryad.org/stash/share/Yke7zt5MuVeD7aE8ie5G_jrbYPE8ZaRCLH58FuYI9QI).

We thank the editor for this comment, we have removed the phrase data not shown.

Reviewer #1

We thank the reviewer for the positive review of our manuscript. The reviewer has made some critical and insightful comments that have definitely improved the final version. We have carefully amended the paper as suggested by the reviewer.

Comments 

Although the manuscript titled: Use of medicinal plants for COVID-19 prevention and respiratory symptom treatment during the pandemic in Cusco, Peru: A cross-sectional survey. is interested and helpful for the people who looks for natural treatment to reduce the risk of COVID-19, but there are some points should be consider before publication:

1. Please be specific in your explanation and comparison of the doses used for each medicinal plant to prevent and treat COVID-19.

We thank the reviewer for this comment, our study had the objective to determine the association between the use of medicinal plants and the treatment or prevention of respiratory symptoms in the population of the five districts of Cusco, Peru. The people in Peru consume these medicinal plants as the fresh form (i.e. ginger) or dry leaves, both in infusions with hot water. Therefore, a dose of medicinal plants could not be determined because, as observed in the survey, the consumption question of medicinal plants was dichotomous (yes/no). We did not assess frequency of consumption and we did not use dietary supplements that were standardized in dose. We are working on another study where we will assess frequency, doses and compare the consumption of medicinal plants in its native state and as standardized dietary supplements. We have updated the limitations section to reflect these observations.

2. The authors did not specify which medicinal plant was effective or very effective in preventing or treating COVID-19; please be specific in your explanation.

We thank the reviewer for this comment, our study did not assess the effectiveness of the medicinal plants in preventing or treating COVID-19. We aimed to determine what plants the surveyed population used during the pandemic and for what symptoms. However, we did not assess symptom improvement or effectiveness against COVID-19 prevention or treatment.

3. The authors use a combination of medicinal plants to prevent or treat COVID-19; perhaps the combination has a greater effect than a single plant; please be specific in your explanation.

We thank the reviewer for this comment, we did not use combination of medicinal plants. We asked the respondents to select from a list of 17 selected medicinal plants the ones they have used to prevent or treat COVID-19 related respiratory symptoms. Therefore, we did not measure efficacy nor effectiveness.

4. Please specify whether the authors used oral, injection, or smelling the steam of hot water from the medicinal plant for prevention or treatment.

We thank the reviewer for this comment, the people in Peru consume these medicinal plants as the fresh form (i.e. ginger) or dry leaves, both in infusions with hot water. We have added this in the discussion section. However, we did not ask what type of preparation they used.

5. The authors should write a good conclusion and be specific in their evaluation of medicinal plants.

We thank the reviewer for this comment, we have improved our conclusion.

6. Editing and correction of the manuscript in English.

We thank the reviewer for this comment, we have proofread the entire manuscript and made editorial corrections to improve clarity.

Reviewer #2

We thank the reviewer for the positive review of our manuscript. The reviewer has made some critical and insightful comments that have definitely improved the final version. We have carefully amended the paper as suggested by the reviewer.

Comments 

The manuscript entitled " Use of medicinal plants for COVID-19 prevention and respiratory symptom treatment during the pandemic in Cusco, Peru: A cross-sectional survey". Title, abstract and overall rationale of work to some extent is good. However, there are still some minor concerns, which needs to be addressed minor revision.

1. The content of the manuscript is too short, specifically introduction and material methods section. Moreover, mechanism section is not properly elaborated, which is important for the reproducibility of the research.

We thank the reviewer for this comment, we have added additional information in the introduction section. We have also expanded the Materials and Methods section; we have included the survey in English and in the original language (Spanish). Furthermore, we have included the raw data of the study in data repository for access and reproducibility.

2. Authors have mentioned very little about COVID-19 in introduction. Without the start of fundamentals about the subject in any manuscript doesn’t provide understanding to all types of readers. Therefore, it should be reader friendly with a proper flow. Authors can use below mentioned references, which will help them in adding this paragraph in introduction and discussion section and can be cited: 10.1039/d0ra06379g, 10.1080/07391102.2020.1802345, 10.3390/plants9091244, 10.1155/2020/8835986, 10.1007/s11356-020-12165-1, 10.1007/s12035-021-02318-9.

We thank the reviewer for this comment, we have added the mentioned references to enhance the introduction and discussion sections.

3. I would suggest the authors to enhance your theoretical discussion and arrives your debate or argument.

We thank the reviewer for this comment, we have enhanced our discussion.

4. Conclusion section must be elaborated and I highly suggest author to write future prospective.

We thank the reviewer for this comment, we have added topics to include in further studies.

Reviewer #3

We thank the reviewer for the positive review of our manuscript. The reviewer has made some critical and insightful comments that have definitely improved the final version. We have carefully amended the paper as suggested by the reviewer.

Comments 

Since the beginning of the COVID-19 epidemic, people around the world have been under constant stress. Currently, there is no established pharmacological strategy for the prevention and/or treatment of a new coronavirus infection. In this situation, we only have to wait for the vaccine and strengthen immunity to keep ourselves and our loved ones healthy. The topic touched upon in the article is relevant. The scientific content of the manuscript justifies its publication. 

1. The authors should justify the list of used medicinal plants in the questionnaire.

We thank the reviewer for this comment, we have added in the methods section the published ethnopharmacological references we used to select the plants used for respiratory problems in Peru.

Reviewer #4

We thank the reviewer for the positive review of our manuscript. The reviewer has made some critical and insightful comments that have definitely improved the final version. We have carefully amended the paper as suggested by the reviewer.

Comments 

The manuscript with the title "Use of medicinal plants for COVID-19 prevention and respiratory symptom treatment during the pandemic in Cusco, Peru: A cross-sectional survey" describes and analyzed some data to know some factors associated with the use of medicinal plants as an adjuvant for Covid19 treatment or prevention. Several analyses have shown the correlation of the medicinal plant consumption for the covid-19 and specific distinct population in Cusco, Peru. The authors claimed that the present article is the first investigation and could become a basis for other future studies. The article is well written. However, several issues need to be resolved:

1. The authors should mention the reason why they select 17 medicinal plants in this study

We thank the reviewer for this comment, we have added in the methods section the published ethnopharmacological references we used to select the plants used for respiratory problems in Peru.

2. The authors showed the respondent experience as Covid-19 patients (table 2); however, there was no data to confirm it. Please explain this condition.

We thank the reviewer for this comment, there are no restrictions for the data. We have uploaded the anonymized data set at the Dryad data repository, and have modified the Supporting Information statement to: Anonymized data set supporting the findings of this study is stored at the Dryad data repository (https://datadryad.org/stash/share/Yke7zt5MuVeD7aE8ie5G_jrbYPE8ZaRCLH58FuYI9QI).

3. The data showed specific medicinal plants for the treatment Covid-19 (from each medicinal plant). Showing the combination of some medicinal plants for treating respiratory symptoms has become more interesting.

We thank the reviewer for this comment, we asked the respondents to select from a list of 17 selected medicinal plants the ones they have used to prevent or treat COVID-19 related respiratory symptoms. We agree that the combination of plants would have bene an interesting approach, but first we wanted to assess which of the 17 plants were used for treating respiratory symptoms. We have added in the conclusions that: Further studies are warranted to determine proper doses, forms of preparation and potential combination of these medicinal plants.

4. Several articles mention that some medicinal plants have a potential effect for Covid-19 treatment or prevention. However, the data of adverse effects from herbal medicine also cannot be excluded. The authors should raise this issue in the discussion. You can see this reference https://doi.org/10.1155/2020/2560645.

We thank the reviewer for this comment, we have added in the discussion the potential toxic effects of asmachilca, echinanacea and cinchona tree.

Reviewer #5

We thank the reviewer for the positive review of our manuscript. The reviewer has made some critical and insightful comments that have definitely improved the final version. We have carefully amended the paper as suggested by the reviewer.

Comments 

GENERAL COMMENTS

1. Language and style should be checked and improved.

We thank the reviewer for this comment, we have proof read the entire manuscript and have adjusted the style to the journal requirements.

2. Exact botanical names should be provided (www.theplantlist.org ) 

We thank the reviewer for this comment, we have verified and corrected the botanical names where appropriate.

3. In Material and methods. What were the criteria used to include or exclude participants after the web-based investigation”?

We thank the reviewer for this comment. Residents of Cusco were included during the period the survey was performed: August and September 2020), who agreed to participate in the research and who were of legal age. There were no exclusions because we implemented that it was mandatory to reply all the answers. Therefore, we only obtained complete and high-quality answers, which was verified by a data quality check. We have added these statements in the Materials and Methods section.

4. A diagram about the design of the study would add an important value to the paper. Please insert the number of participants in each step. 

We thank the reviewer for this comment. However, by not having exclusion criteria, the total number of respondents corresponded to the number of completed surveys. Therefore, the requested diagram was not included.

5. How the authors calculated the sample? I need more details to judge the process. 

We thank the reviewer for this comment, the Study Design subsection details how the sample size was calculated: The pilot data was used to calculate the minimal sample size necessary for the actual study. It was determined that a minimum sample size of 1,530 was necessary to achieve a minimum percentage difference of 2.5% (49.0% versus 51.5%), a statistical power of 80%, and a confidence level of 95%. The sample size was calculated using power analysis. 

6. The selection seems to be biased. I need the exact process of participants’ selection.

We thank the reviewer for this comment. The reviewer is correct that there is a selection bias, by not having performed random sampling to obtain the responses. It needs to be understood that during the time the survey was performed, there were still strict social isolation measurements in Peru. Thus, in order to protect the research team, we performed an online survey. Regardless, we were able to collect a large sample size of responses, which allowed us for a proper statistical analysis. We have added selection bias as another limitation in the manuscript.

7. In the study protocol, participants were asked to select from a list of selected 17 medicinal species. This seems to be a serious bias. The authors should clarify and justify this approach.

We thank the reviewer for this comment, we asked the respondents to select from a list of 17 selected medicinal plants the ones they have used to prevent or treat COVID-19 related respiratory symptoms. We have added in the methods section the published ethnopharmacological references we used to select the plants used in the survey. Peru has hundreds of medicinal plants used for respiratory symptoms, having an open question could have diluted the number of responses and complicated the statistical analysis.

8. In the section results and discussion, replace the names of plants by their scientific names. 

We thank the reviewer for this comment, we have added the scientific names in the Results and Discussion sections.

9. For the mentioned species, important details are missing (parts used, administration routes, preparation, …..). This important information should be added. 

We thank the reviewer for this comment, we did not ask to specify what part of the medicinal plant is consumed, nor the preparation, nor the administration route. The people in Peru consume these medicinal plants as the fresh form (i.e. ginger) or dry leaves, both in infusions with hot water. We have specified this in the manuscript.

SPECIFIC COMMENTS

Section. Introduction. 

1. The paragraph “The idiosyncrasy of the population has also focused on the use of medicinal plants, natural products …………….. in the context of the COVID19 pandemic.” Remove this paragraph and replace it with an overview about recent original or review papers discussing the use of medicinal plants to treat COVID-19. 

We thank the reviewer for this comment, we have added in the Discussion information about published papers and reviews related to the use of medicinal plants related to COVID-19. We decided to added into the Discussion since it fits better in that section.

Section. M&M. 

2. “The respondents were asked to indicate if during the COVID-19 lockdown …………….. Center of Disease Control and Prevention (CDC) (45)” This paragraph should be removed since it reports more details atht can be considered of less importance for the reader. 

We thank the reviewer for this comment. However, we respectfully disagree in removing that paragraph. We added those details so that the reader can understand the reasoning behind the order of questions in the survey and the reasoning behind it.

3. “The respondents were asked to select from a list of 17 selected medicinal plants the ones they have used to prevent or treat COVID-19 related respiratory symptoms.” Why the authors asked the participants to select from a pre-established list of species? It seems that this can be a serious bias.

We thank the reviewer for this comment, we asked the respondents to select from a list of 17 selected medicinal plants the ones they have used to prevent or treat COVID-19 related respiratory symptoms. We have added in the methods section the published ethnopharmacological references we used to select the plants used in the survey. Peru has hundreds of medicinal plants used for respiratory symptoms, having an open question could have diluted the number of responses and complicated the statistical analysis.

Section. Results 

4. The associations between gender and different parameters should be discussed carefully since female were predominant. It’d be an effect of dominance and not a real effect of gender. Page 14. “Therefore, it could be observed that in general the respondents preferably used medicinal plants as treatment of respiratory symptoms.” I think this sentence can lead the reader to a wrong conclusion. This should be removed and discussed in the discussion section. 

We thank the reviewer for this comment, we have removed the sentence and have improved the discussion section.

Section. Discussion 

5. The most cited species should be further discussed and their use against COVID-19 should justified (bioactive compounds, previous in vitro or in vivo studies, ethnobotanical investigations especially in the region reporting their use to treat respiratory diseases, …).

We thank the reviewer for this comment, we have included those details and had overall improved the discussion section.

6. Page 14. “As shown on Table 3, the respondents mentioned the use of 17 medicinal plants for the treatment of COVID-19 related respiratory symptoms.” By reading the study design, this statement is not true. Participants reported the use form a preselected 17 medicinal species!!!!!!

We thank the reviewer for this comment, we have modified the sentence to: As shown on Table 3, the respondents reported the use of medicinal plants from a preselected list for the treatment of COVID-19 related respiratory symptoms.

---

## [Editor Report · Decision Letter 1]

25 Aug 2021

Use of medicinal plants for COVID-19 prevention and respiratory symptom treatment during the pandemic in Cusco, Peru: A cross-sectional survey

PONE-D-21-15800R1

Dear Dr. Yanez,

We’re pleased to inform you that your manuscript has been judged scientifically suitable for publication and will be formally accepted for publication once it meets all outstanding technical requirements.

Kind regards,

Mohd Adnan, PhD

Academic Editor

PLOS ONE
---

## [Editor Report · Acceptance letter]

14 Sep 2021

PONE-D-21-15800R1 

Use of medicinal plants for COVID-19 prevention and respiratory symptom treatment during the pandemic in Cusco, Peru: A cross-sectional survey 

Dear Dr. Yañez:

I'm pleased to inform you that your manuscript has been deemed suitable for publication in PLOS ONE. Congratulations! Your manuscript is now with our production department. 

Kind regards, 

on behalf of

Dr. Mohd Adnan 

Academic Editor

PLOS ONE